# Efficacy and safety of decompressive craniectomy with non-suture duraplasty in patients with traumatic brain injury

**Tae Seok Jeong**, **Gi Taek Yee** *, **Tae Gyu Lim, Woo Kyung Kim**, **Chan Jong Yoo**

Department of Neurosurgery, Gachon University Gil Medical Center, Incheon, Korea

☯ These authors contributed equally to this work.

* gtyee@gilhospital.com

## Abstract

### Background

Decompressive craniectomy is an important surgical treatment for patients with severe traumatic brain injury (TBI). Several reports have been published on the efficacy of non-watertight sutures in duraplasty performed in decompressive craniectomy. This study sought to determine the safety and feasibility of the non-suture dural closure technique in decompressive craniectomy.

### Methods

A total of 106 patients were enrolled at a single trauma center between January 2017 and December 2018. We retrospectively collected data and classified the patients into non-suture and suture duraplasty craniectomy groups. We compared the characteristics of patients and their intra/postoperative findings such as operative time, blood loss, imaging findings, complications, and Glasgow Outcome Scale scores.

### Results

There were 37 and 69 patients in the non-suture and suture duraplasty groups, respectively. There were no significant differences between the two groups concerning general characteristics. The operative time was significantly lower in the non-suture duraplasty group than in the suture duraplasty group (150 min vs. 205 min; p = 0.002). Furthermore, blood loss was significantly less severe in the non-suture duraplasty group than in the suture duraplasty group (1000 mL vs. 1500 mL; p = 0.028). There were no other significant differences.

### Conclusion

Non-suture duraplasty involved shorter operative times and less severe blood losses than suture duraplasty. Other complications and prognoses were similar across groups. Therefore, the non-suture duraplasty in decompressive craniectomy is a safe and feasible surgical technique.

**Data Availability Statement:** All relevant data are within the paper and its Supporting Information files.

**Funding:** The authors received no specific funding for this work.

**Competing interests:** The authors have declared that no competing interests exist.

## Introduction

Decompressive craniectomy is a neurosurgical procedure used for lowering the increased intracranial pressure in patients with severe traumatic brain injury (TBI). However, there is no established standard guideline for performing decompressive craniectomy. Guresir et al. [1] reported that rapid closure decompressive craniectomy without duraplasty was a safe and feasible method for the management of malignant brain swelling. The dura closure technique is mostly dependent on the clinician's experience. The dura suturing technique is traditionally known to require watertight closure to prevent complications such as cerebrospinal fluid (CSF) leakage and infection. Several studies have reported that watertight duraplasty should not be used in decompressive craniectomy because non-watertight duraplasty can reduce the operative time while the probability of complications remains the same [1–3]. In cases of TBI, few reports have been carried out on duraplasty without suturing. Therefore, the purpose of this study was to determine the safety and feasibility of non-suture duraplasty in the context of decompressive craniectomy in TBI patients.

## Materials and methods

### Patient population

This retrospective study was approved by our institutional review board (GCIRB2020-088). The requirement for obtaining informed consent from the patients was waived because the study was based on the information collected as part of routine clinical care and medical record-keeping. We reviewed the data of 151 TBI patients who underwent decompressive craniectomy and were hospitalized in a single trauma center between January 2017 and December 2018. We excluded patients who had previously undergone any surgical treatment for other brain lesions and aged < 18 years or > 70 years. We also excluded cases in which the craniectomy was performed bilaterally or occipitally (posterior fossa), or if an emergency operation for another part of the body was simultaneously performed. Therefore, 106 patients who underwent decompressive craniectomy due to TBI were enrolled in this study and divided into suture duraplasty and non-suture duraplasty groups.

### Data analysis

Data on general characteristics such as sex, age, and diagnosis, and pertaining to trauma such as the Glasgow Coma Scale (GCS), anisocoria, and injury severity score (ISS) were retrospectively reviewed. We also reviewed intra/postoperative findings such as operative time, blood loss, and computed tomography (CT) scans; complications such as wound dehiscence and surgical site infection (SSI); and prognostic indicators such as the Glasgow Outcome Scale (GOS). In addition, the results of cranioplasty, including intraoperative findings, GOS, and complications, were reviewed, although the limited nature of the data meant that it could not be thoroughly evaluated.

To quantify blood loss, an anesthesiologist measured the amount of fluid, including blood and irrigation fluid, collected through suction during surgery and then subtracted the amount of fluid used for irrigation from the measured volume. Final blood loss levels were also calculated based on the amount of blood on the gauze, according to a method proposed by Ali Algadiem et al. [4].

Wound dehiscence was defined when a surgical incision reopened, without any discharge such as in the form of pus, and when bacteria were not identified from the opened site to distinguish superficial incisional SSIs. When CSF was drained through the surgical wound, it was determined whether it was a wound dehiscence or SSI, depending on the results of the culture

test. When CSF was drained to the subcutaneous/epidural space, but not through the surgical wound, it was diagnosed as a subgaleal fluid collection.

The SSI was defined based on the US Centers for Disease Control and Prevention (CDC) criteria [5, 6].

## Surgical procedures

The standard decompressive craniectomy technique included bone removal, dural incision, and hematoma removal. A star-shaped dural incision was made (Fig 1). Duraplasty was conducted in two different ways, with and without sutures. Suture duraplasty was performed with watertight suturing of the dura with the artificial dura. Non-suture duraplasty was performed by overlaying an artificial dura over the dural opening area, without any suturing (Fig 2). The choice of surgical technique (suture or non-suture duraplasty) depended on the surgeon's preference. After duraplasty, the subcutaneous tissue and skin were closed in a serial fashion.

## Statistical analysis

Continuous data (e.g., age) were presented as medians and interquartile ranges. Categorical data (e.g., sex) were presented as frequencies and percentages. Continuous variables were compared using the independent t-test, and categorical variables were compared using Fisher's

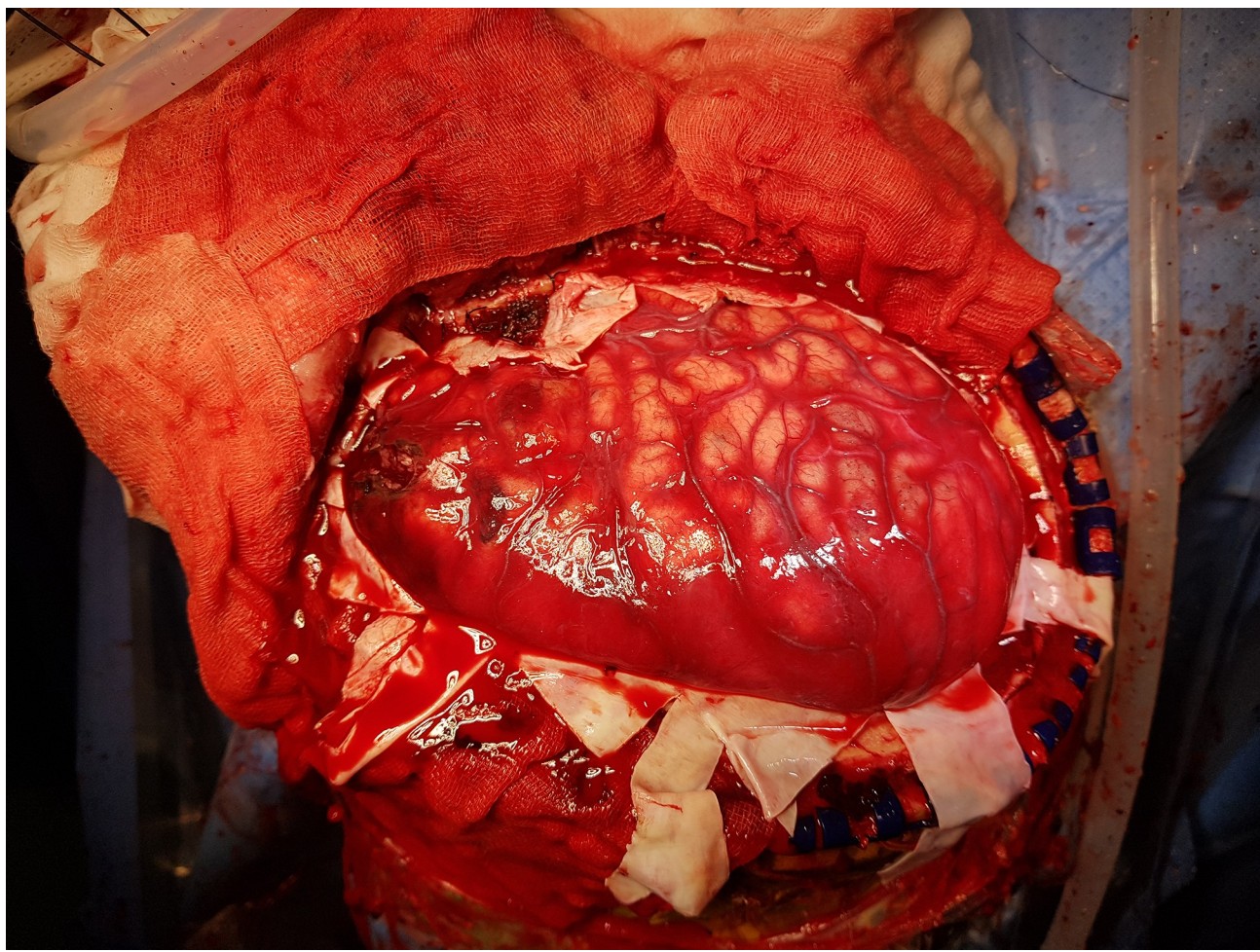

**Fig 1. Intraoperative photograph showing wide craniotomy and dura opening.**

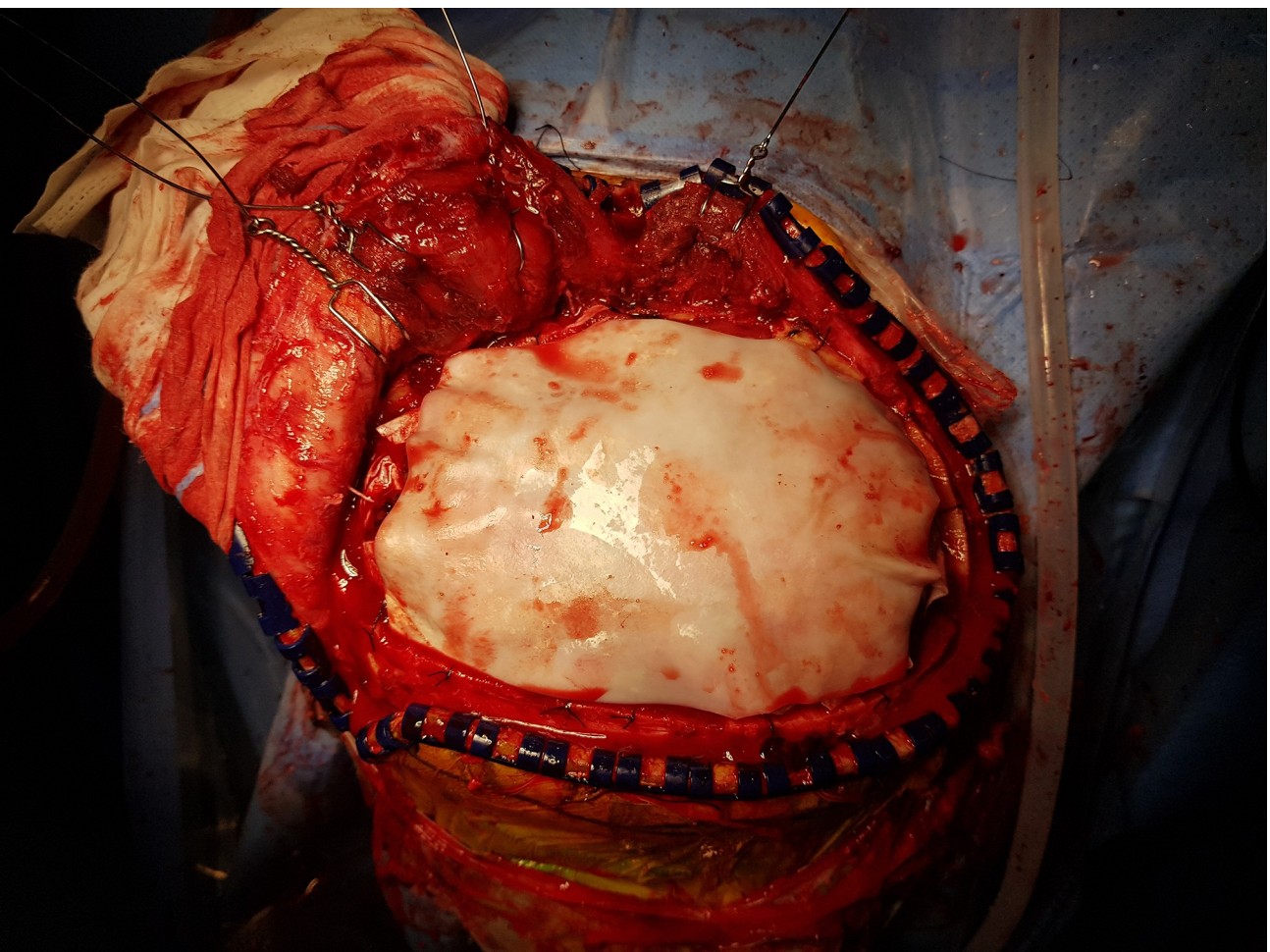

**Fig 2. Intraoperative photograph showing non-suture duraplasty with an artificial dura material overlaid on the brain cortex.**

exact or Pearson's chi-square test. All tests were performed using a statistical significance criterion of α = 0.05, and analyses were performed in SPSS for Windows version 23.0 (IBM Corp., Armonk, NY, USA).

## Results

### Patient characteristics

Of the 106 patients (female: 31, male: 75), 69 belonged to the suture duraplasty group and 37 to the non-suture duraplasty group. The median age of the subjects was 53 years, and the most common diagnosis was that of a subdural hematoma (76.4%). The GCS score of the subjects on admission was classified into three categories according to TBI severity: mild (GCS: 13~15), moderate (9~12), and severe (3~8). Severe TBIs (67.9%) were the most common. The median ISS was 25. There were no significant differences between the two groups with respect to patient characteristics (Table 1).

### Intra/Postoperative findings

As intraoperative findings, operative time and blood loss were analyzed. The operative time was 205 min for the suture duraplasty group and 150 min for the non-suture duraplasty group;

**Table 1. Patient characteristics.**

| | Suture duraplasty (n = 69) | Non-suture duraplasty (n = 37) | Total (n = 106) | *p*-value |
|---|---|---|---|---|
| Sex | | | | 0.597 |
| Female | 19 (27.5) | 12 (32.4) | 31 (29.2) | |
| Male | 50 (72.5) | 25 (67.6) | 75 (70.8) | |
| Age (years) | 52 (42–60) | 56 (43–59) | 53 (42–60) | 0.942 |
| Diagnosis | | | | 0.269 |
| Subdural hematoma | 55 (79.7) | 26 (70.3) | 81 (76.4) | |
| Intracerebral contusion/hematoma | 14 (20.3) | 11 (29.7) | 25 (23.6) | |
| GCS score | | | | 0.723 |
| 3–8 | 48 (69.6) | 24 (64.9) | 72 (67.9) | |
| 9–12 | 9 (13.0) | 7 (18.9) | 16 (15.1) | |
| 13–15 | 12 (17.4) | 6 (16.2) | 18 (17.0) | |
| Anisocoria | | | | 0.853 |
| Yes | 33 (47.8) | 17 (45.9) | 50 (47.2) | |
| No | 36 (52.2) | 20 (54.1) | 56 (52.8) | |
| ISS | 25 (17–30) | 25 (24–33) | 25 (21–33) | 0.152 |

GCS: Glasgow Coma Scale; ISS: Injury Severity Score. Values are presented as median values (interquartile ranges) or numbers (%).

the non-suture duraplasty group had a significantly shorter operative time (p = 0.002). In addition, blood loss was significantly less severe in the non-suture duraplasty group (1000 mL) than in the suture duraplasty group (1500 mL, p = 0.028).

A postoperative brain CT scan was done one month after surgery. The amount of subgaleal fluid collection did not differ between the two groups, and ipsilateral/contralateral subdural hygromas tended to be more severe in the suture duraplasty group; however, there was no statistically significant difference.

Wound dehiscence and SSIs were evaluated for the presence of any complications, revealing no significant differences between the two groups. GOS scores at discharge also did not differ between the two groups (Table 2).

## Cranioplasty

We observed 37 patients (suture duraplasty group: 27, non-suture duraplasty group: 10) who underwent cranioplasty following a decompressive craniectomy. GOS scores did not differ between the two groups. The incidence of complications, including infection after cranioplasty, were similar in the suture duraplasty and non-suture duraplasty groups (Table 3). In the non-suture duraplasty craniectomy group, the cortex was not exposed because of the formation of fibrotic tissue between the scalp flap and dura material (Fig 3).

## Discussion

It is normal for individual differences to exist with regard to the dura reconstruction method performed by different surgeons, as these vary based on the training received. Traditionally, many surgeons recommend tightly closing the dura mater (i.e., 'watertight suture') because this has been thought to prevent various complications, such as infection, CSF leakage, and subgaleal fluid collection. However, the efficacy of watertight suturing remains unknown, especially in TBI cases. Recently, some authors have argued that watertight suturing increases operative time and hospital costs, and the occurrence of complications is similar to that in

**Table 2. Intra/postoperative findings and results.**

|  | Suture duraplasty (n = 69) | Non-suture duraplasty (n = 37) | *p*-value |
|---|---|---|---|
| Intraoperative findings |  |  |  |
| Operative time (min) | 205 (160–265) | 150 (105–225) | 0.002* |
| Blood loss (mL) | 1500 (1000–2500) | 1000 (700–1800) | 0.028* |
| Post-operative CT findings |  |  |  |
| Subgaleal fluid collection | 12 (17.4) | 9 (24.3) | 0.648 |
| Ipsilateral subdural hygroma | 16 (23.2) | 4 (10.8) | 0.121 |
| Contralateral subdural hygroma | 12 (17.4) | 2 (5.4) | 0.131 |
| Complications |  |  |  |
| Wound dehiscence | 10 (14.5) | 4 (10.8) | 0.766 |
| Surgical site infection | 6 (8.7) | 2 (5.4) | 0.710 |
| GOS score |  |  | 0.834 |
| 1 | 9 (13.0) | 6 (16.2) |  |
| 2 | 12 (17.4) | 9 (24.3) |  |
| 3 | 25 (36.2) | 13 (35.1) |  |
| 4 | 17 (24.6) | 7 (18.9) |  |
| 5 | 6 (8.7) | 2 (5.4) |  |

CT: Computed Tomography; GOS: Glasgow Outcome Scale. Values are presented as median values (interquartile ranges) or numbers (%).

*Statistically significant differences (*p*-value <0.05).

cases without watertight sutures [1, 2, 7]. Guresir et al. [1] introduced a rapid closure decompressive craniectomy, which is not a dura reconstruction, but rather fills the dural defect with a hemostat material. Non-suture duraplasty craniectomy is similar to a rapid closure craniectomy but uses synthetic dura instead of the hemostat material to avoid adhesion in cranioplasty. This difference may incur higher costs for non-suture duraplasty craniectomies than rapid closure craniectomies. However, there may be surgical benefits, such as shortened operative times, because of inhibited adhesion in the cranioplasty after non-suture duraplasty craniectomy.

This study assessed the operative time and blood loss as comparable intraoperative parameters. The non-suture duraplasty craniectomy group was found to have a shorter operative time and less severe blood loss than the suture group. It is known that shorter operative times and

**Table 3. Complications and GOS after cranioplasty.**

|  | Suture duraplasty (n = 27) | Non-suture duraplasty (n = 10) |
|---|---|---|
| Complications |  |  |
| Surgical site infection | 1 (3.7) | 0 (0) |
| Postoperative epidural hematoma | 1 (3.7) | 1 (10) |
| GOS score |  |  |
| 1 | 0 (0) | 0 (0) |
| 2 | 0 (0) | 0 (0) |
| 3 | 16 (59.3) | 6 (60) |
| 4 | 7 (25.9) | 2 (20) |
| 5 | 4 (14.8) | 2 (20) |

GOS: Glasgow Outcome Scale. Values are presented as numbers (%).

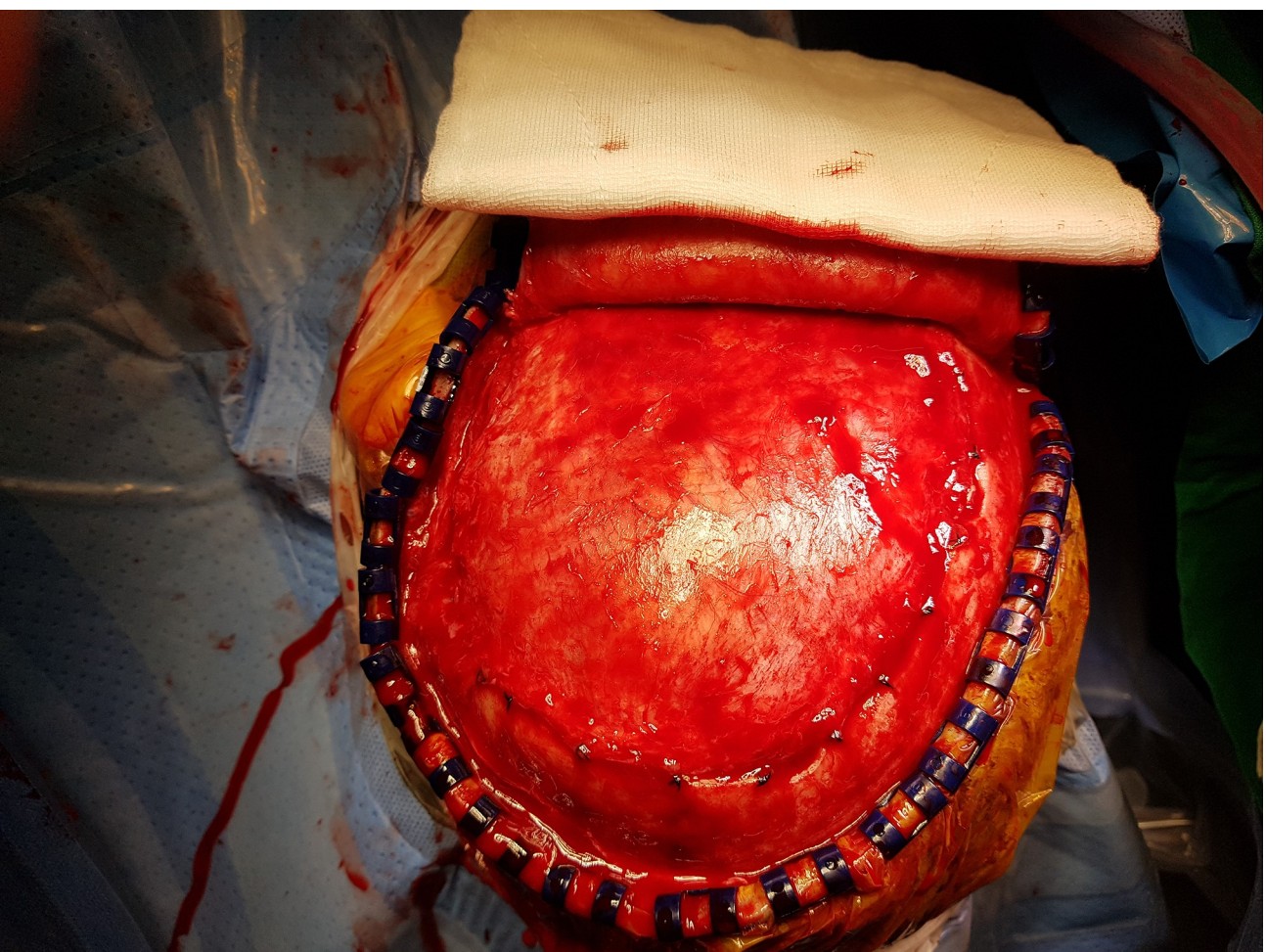

**Fig 3. Intraoperative photograph showing craniectomy site covered by fibrotic tissue in cranioplasty following decompressive craniectomy with non-suture duraplasty.**

less severe bleeding levels help to minimize the possibility of infection and hospital costs [8, 9]. It may also help treat patients requiring critical care, such as for multiple trauma [10, 11].

The incidence of postoperative wound infections documented in the literature ranges from as low as 1.25% to as high as 17% without prophylactic antibiotics, and 0.3% to 3.0% with prophylactic antibiotics [10–14]. The SSI incidence in the present study was approximately 7.5%, which is slightly higher than that in previous studies. Korinek et al. [10] reported that independent predictive risk factors of SSI after craniotomy included a recent neurosurgical operation, contaminated wounds, prolonged operative times, and having had an emergency operation. A slightly higher infection rate in this study was attributed to the fact that all patients included in the study underwent emergency surgery. In this study, the SSI incidence of the non-suture duraplasty craniectomy group was lower than that of the suture group, although there was no statistical significance. This may be because the operative time of the non-suture duraplasty craniectomy group was shorter than that of the suture group.

All cases of wound dehiscence were treated with a simple repair. The SSI was initially treated with antibiotics, which were selected according to the wound culture results. When the wound opens and pus is discharged, or when inflammation is not controlled despite the use of antibiotic therapy, surgical wound revision is carried out. In our study, four cases of suture

duraplasty and two cases of non-suture duraplasty were treated with antibiotics. In addition, two cases of suture duraplasty were treated with antibiotics and surgical revision, including irrigation and granulation tissue removal, because pus was discharged from the opened wound.

The incidence of subdural hygroma after decompressive craniectomy in TBI patients was reported to be 21–50% [7, 15–18]. Subdural hygroma frequently occurs on the ipsilateral side of the craniectomy; however, it can occur on the contralateral side or bilaterally. Surgical treatment is not always required but is necessary if subdural hygroma causes a mass effect and becomes symptomatic [16, 17]. In this study, the incidence of subdural hygroma (both ipsilateral and contralateral) was about 32.1% (similar to previous studies) and was less severe in the non-suture duraplasty craniectomy group than the suture group. If watertight duraplasty is performed, even in conjunction with decompressive duraplasty, the brain may still be compressed by the dura or subdural fluid because the progression of brain edema cannot be predicted. In cases of insufficient decompressive duraplasty, subdural hygroma is more likely to occur because of a decrease in the infusion of CSF circulation and a brain shift [7]. Therefore, non-suture duraplasty craniectomy may help reduce the incidence of subdural hygroma, even in cases of severe brain edema, because there is no compression from the dura. However, the above results stem from preliminary reports with insufficient statistical evidence.

Subgaleal fluid collection was somewhat higher in the non-suture duraplasty group. Subgaleal fluid collection is thought to be mainly caused by CSF leaking out of the dura. It can cause trouble when fluid collection increases and incurs a massive effect, or when it works as a source of the SSI. Vieira et al. [3] reported that once the arachnoid is intact, there is no increased risk of CSF leakage, and watertight closures may lead to small defects on suture lines, causing a "one-way valve" effect that may potentially facilitate the development of CSF leakage. In suture duraplasty, CSF can be collected out of the dura unidirectionally using a one-way valve effect and can have a massive effect. However, in non-suture duraplasty, CSF is less likely to cause a massive effect by moving in and out of the dura. In patients who underwent decompressive craniectomy, it has been observed that the scalp of the craniectomy site swells up after coughing and gradually sinks as CSF moves in and out of the dura. In our study, only one brain CT scan per month after surgery was analyzed, meaning that only the condition at that time was effectively evaluated. It is necessary to evaluate the patterns of change of the subgaleal fluid collection through serial imaging analysis. Concerning the source of the SSI, subgaleal fluid collection is not likely to increase the SSI incidence because the SSI rate in the non-suture duraplasty group was lower than that of the suture duraplasty group.

In our study, all cases of subdural hygroma were observed without specific treatments because there were no findings linked to a massive effect or neurological symptoms caused by subdural hygroma, and most subdural hygromas resolved spontaneously. In addition, most patients with subgaleal fluid collection were closely observed because this did not incur a massive effect and was sometimes spontaneously absorbed. However, in two patients from the suture duraplasty group, subgaleal fluid collection increased gradually and compressed the brain, such that CSF leakage points were surgically repaired.

Although there were not many cases, the results of cranioplasty showed that there was no significant difference between the two groups in GOS and complications. Therefore, we believe that non-suture duraplasty may not cause any problems in cranioplasty, although further evaluation is warranted.

This study has some limitations. The data were analyzed retrospectively, and the sample size was relatively small. In addition, a decompressive craniectomy was performed by multiple surgeons. This may have influenced intra/postoperative results such as operative time, blood loss, and prognosis.

Despite these limitations, this study's results may provide useful information with regard to the benefits of non-suture duraplasty, such as shorter operative time and less severe blood loss, during decompressive craniectomy.

## Conclusions

This study was a report on non-suture duroplasty performed as part of decompressive craniectomy in TBI patients. Non-suture duraplasty revealed a shorter operative time and less severe blood losses than suture duraplasty. Other complications and prognoses were similar in both groups. Therefore, it can be assumed that non-suture duraplasty in decompressive craniectomy is a safe and feasible technique.

## Supporting information

**S1 Data.**
(XLSX)

## Acknowledgments

We thank Dae Han Choi for technical work on the graphics.

## Author Contributions

**Conceptualization:** Tae Seok Jeong, Gi Taek Yee, Tae Gyu Lim.

**Data curation:** Tae Seok Jeong, Tae Gyu Lim.

**Formal analysis:** Tae Seok Jeong.

**Investigation:** Tae Seok Jeong.

**Methodology:** Tae Seok Jeong, Tae Gyu Lim.

**Project administration:** Tae Seok Jeong.

**Resources:** Tae Seok Jeong, Gi Taek Yee, Woo Kyung Kim, Chan Jong Yoo.

**Software:** Tae Seok Jeong.

**Supervision:** Tae Seok Jeong, Gi Taek Yee.

**Validation:** Tae Seok Jeong, Gi Taek Yee, Woo Kyung Kim, Chan Jong Yoo.

**Visualization:** Tae Seok Jeong.

**Writing – original draft:** Tae Seok Jeong.

**Writing – review & editing:** Tae Seok Jeong, Gi Taek Yee.

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
