## [Decision Letter · Decision Letter 0]

13 May 2020

PONE-D-20-10922

Efficacy and safety of decompressive craniectomy with non-suture duraplasty in patients with traumatic brain injury

PLOS ONE

Dear Dr. Yee,

Thank you for submitting your manuscript to PLOS ONE. After careful consideration, we feel that it has merit but does not fully meet PLOS ONE’s publication criteria as it currently stands. Therefore, we invite you to submit a revised version of the manuscript that addresses the points raised during the review process.

The manuscript should address all the concerns raised during the revision process. 

We would appreciate receiving your revised manuscript by Jun 27 2020 11:59PM. To enhance the reproducibility of your results, we recommend that if applicable you deposit your laboratory protocols in protocols.io, where a protocol can be assigned its own identifier (DOI) such that it can be cited independently in the future. For instructions see: http://journals.plos.org/plosone/s/submission-guidelines#loc-laboratory-protocols

We look forward to receiving your revised manuscript.

Kind regards,

Giovanni Grasso

Academic Editor

PLOS ONE

Journal Requirements:

Reviewers' comments:

Reviewer's Responses to Questions

**Comments to the Author**

1. Is the manuscript technically sound, and do the data support the conclusions?

Reviewer #1: Partly

2. Has the statistical analysis been performed appropriately and rigorously? 

Reviewer #1: Yes

3. Have the authors made all data underlying the findings in their manuscript fully available?

Reviewer #1: Yes

4. Is the manuscript presented in an intelligible fashion and written in standard English?

Reviewer #1: No

5. Review Comments to the Author

Reviewer #1: First of all, I would like to congratulate the authors for their work.

There are some issues, however, that need clarification and further explanation.

#1: on page 2, line 30: “There were 37 patients in the non-suture group and 69 in the suture craniectomy group” – I believe the sentence should be: “There were 37 patients in the non-suture duraplasty and 69 in the suture duraplasty group.”

#2: on page 4, line 52: “Therefore, the purpose of this study was to determine if non-watertight dura reconstruction contributes significantly to the success of decompressive craniectomy in TBI” – I believe both techniques (with and without watertight dural closure) may lead to a successful decompressive craniectomy. The focus should be on the safety and feasibility of the non-watertight dural closure procedure.

#3: on page 5: the authors should clarify the definition of wound dehiscence. Also it is important to explain how the intraoperative blood loss was measured. Furthermore, the most feared complication of a non-watertight duraplasty decompressive craniectomy , CSF leak, is not analyzed. If possible, the authors should include these data on their work.

#4: on page 7: the treatment of epidural hematomas usually involves craniotomy, hematoma evacuation, dural tenting sutures and fixation of the bone flap. In the majority of the cases, it does not involve duraplasty and/or decompressive craniectomy. Yet, almost 15% of the patients included in the study had an epidural hematoma. The authors need to clarify why decompressive craniectomy was performed in this subset of patients.

# 5: on page 8: it is important to clarify how complications (subgaleal fluid collections, hygromas, wound dehiscence and surgical site infections) were treated (if any treatment was needed at all).

# 6: on page 10, line 159: “Surgical treatment is not always required, but may be necessary if it converts to a chronic subdural hematoma or is caused by a mass effect”- I believe hygromas should be treated if it causes mass effect.

# 7: on page 10: in nearly 35% of the patients cranioplasty was performed. I believe it would be more appropriate to include this data in the methods and results sections rather than just in the discussion.

6. PLOS authors have the option to publish the peer review history of their article (what does this mean?). If published, this will include your full peer review and any attached files.

Reviewer #1: Yes: Eduardo Vieira

---

## [Author Response · Author response to Decision Letter 0]

9 Jun 2020

Response to Reviewers

Thank you for your sincere review of our paper. I fully agree with what you pointed out and have made several changes to this. The modified part is indicated in red. In addition, English correction was made again, and a certificate for this was attached.

 

#1: on page 2, line 30: “There were 37 patients in the non-suture group and 69 in the suture

craniectomy group” – I believe the sentence should be: “There were 37 patients in the non-suture duraplasty and 69 in the suture duraplasty group.”

It was modified as follows.

There were 37 patients in the non-suture duraplasty and 69 in the suture duraplasty group.

 

#2: on page 4, line 52: “Therefore, the purpose of this study was to determine if non-watertight dura reconstruction contributes significantly to the success of decompressive craniectomy in TBI” – I believe both techniques (with and without watertight dural closure) may lead to a successful decompressive craniectomy. The focus should be on the safety and feasibility of the non-watertight dural closure procedure.

As you pointed out, modifying it to safety and feasibility seems to be more appropriate for the purpose of the study. The technique used in our study was a non-suture duraplasty without suturing, and the contents were modified as follows.

Abstract - Background

“This study sought to determine the safety and feasibility of the non-suture dural closure technique in decompressive craniectomy.”

Introduction

“Therefore, the purpose of this study was to determine the safety and feasibility of non-suture duraplasty in the context of decompressive craniectomy for TBI patients.”

Conclusions

“Therefore, it can be assumed that non-suture duraplasty in decompressive craniectomy is a safe and feasible technique.”

 

#3: on page 5: the authors should clarify the definition of wound dehiscence. Also it is

important to explain how the intraoperative blood loss was measured. Furthermore, the most

feared complication of a non-watertight duraplasty decompressive craniectomy , CSF leak, is

not analyzed. If possible, the authors should include these data on their work.

The definition of wound dehiscence and the method of measuring intraoperative blood loss were added to the method. As you mentioned, we also believe that CSF leak is the most important complication of non-suture duraplasty. In our study, CSF leak was classified as follows. When CSF was discharged out of scalp through the surgical wound, it was classified as wound dehiscence or SSI according to the culture test result, and if CSF was discharged into the subcutaneous/epidural space but was not discharged outside the surgical wound, it was diagnosed as a subgaleal fluid collection. Because the definition of CSF leak may differ from reader to reader, CSF leak has been classified in detail. We have added this to the method. Also, the content of infection and mass effect that CSF leaks can cause has been added to the discussion.

Methods

“To quantify blood loss, an anesthesiologist measured the amount of fluid, including blood and irrigation fluid, collected through suction during surgery and then subtracted the amount of fluid used for irrigation from the measured volume. Final blood loss levels were also calculated based on the amount of blood on the gauze according to a method proposed by Ali Algadiem et al. [4].

Wound dehiscence was defined when a surgical incision reopened, without any discharge such as in the form of pus, and when bacteria were not identified from the opened site to distinguish superficial incisional SSIs. When CSF was drained through the surgical wound, it was determined whether it was a wound dehiscence or SSI, depending on the results of the culture test. When CSF was drained to the subcutaneous/epidural space, but not through the surgical wound, it was diagnosed as a subgaleal fluid collection.”

Discussion

“Subgaleal fluid collection was somewhat higher in the non-suture duraplasty group. Subgaleal fluid collection is thought to be mainly caused by CSF leaking out of the dura. It can cause trouble when fluid collection increases and incurs a massive effect, or when it works as a source of the SSI. Vieira et al. [3] reported that once the arachnoid is intact, there is no increased risk of CSF leakage, and watertight closures may lead to small defects on suture lines, causing a “one-way valve” effect that may potentially facilitate the development of CSF leakage. In suture duraplasty, CSF can be collected out of the dura unidirectionally using a one-way valve effect and can have a massive effect. However, in non-suture duraplasty, CSF is less likely to cause a massive effect by moving in and out of the dura. In patients who underwent decompressive craniectomy, it has been observed that the scalp of the craniectomy site swells up after coughing and gradually sinks as CSF moves in and out of the dura. In our study, only one brain CT scan per month after surgery was analyzed, meaning that only the condition at that time was effectively evaluated. It is necessary to evaluate the patterns of change of the subgaleal fluid collection through serial imaging analysis. With regard to the source of the SSI, subgaleal fluid collection is not likely to increase the SSI incidence because the SSI rate in the non-suture duraplasty group was lower than that of the suture duraplasty group.”

 

#4: on page 7: the treatment of epidural hematomas usually involves craniotomy, hematoma

evacuation, dural tenting sutures and fixation of the bone flap. In the majority of the cases, it

does not involve duraplasty and/or decompressive craniectomy. Yet, almost 15% of the

patients included in the study had an epidural hematoma. The authors need to clarify why

decompressive craniectomy was performed in this subset of patients.

Most traumatic brain injuries do not have only one finding, such as subdural or epidural hematoma, but are accompanied by various hemorrhagic findings. In our study, a representative finding was described as the diagnosis name. As you mentioned, for most epidural hematoma cases of our study, craniotomy without duraplasty was performed. However, if brain swelling/edema by contusions exists or is expected, craniectomy with duraplasty was performed. To clarify the reason for the duraplasty, we modified epidural hematoma in table 1 to epidural hematoma with contusion.

Table 1.

Diagnosis 0.269

 Subdural hematoma 55 (79.7) 26 (70.3) 81 (76.4) 

 Epidural hematoma with contusion 7 (10.1) 8 (21.6) 15 (14.2) 

 Intracerebral hematoma 7 (10.1) 3 (8.1) 10 (9.4) 

 

# 5: on page 8: it is important to clarify how complications (subgaleal fluid collections,

hygromas, wound dehiscence and surgical site infections) were treated (if any treatment was

needed at all).

Thanks for the good point. We have added treatment methods for each complication to the discussion.

Discussion

“All cases of wound dehiscence were treated with a simple repair. The SSI was initially treated with antibiotics, which were selected according to the wound culture results. When the wound opens and pus is discharged, or when inflammation is not controlled despite the use of antibiotic therapy, surgical wound revision is carried out. In our study, four cases of suture duraplasty and two cases of non-suture duraplasty were treated with antibiotics. In addition, two cases of suture duraplasty were treated with antibiotics and surgical revision, including irrigation and granulation tissue removal, because pus was discharged from the opened wound.” 

“In our study, all cases of subdural hygroma were observed without specific treatments because there were no findings linked to a massive effect or neurological symptoms caused by subdural hygroma, and most subdural hygromas resolved spontaneously. In addition, most patients with subgaleal fluid collection were closely observed because this did not incur a massive effect and was sometimes spontaneously absorbed. However, in two patients from the suture duraplasty group, subgaleal fluid collection increased gradually and compressed the brain, such that CSF leakage points were surgically repaired.” 

 

# 6: on page 10, line 159: “Surgical treatment is not always required, but may be necessary if it converts to a chronic subdural hematoma or is caused by a mass effect”- I believe hygromas should be treated if it causes mass effect.

We also believe that surgical treatment is necessary if subdural hygroma causes mass effects and neurological symptoms. We made the following changes.

Discussion

“Surgical treatment is not always required but is necessary if subdural hygroma causes a mass effect and becomes symptomatic.”

Reference. 

Paredes I, Cicuendez M, Delgado MA, Martinez-Pérez R, Munarriz PM, Lagares A. Normal pressure subdural hygroma with mass effect as a complication of decompressive craniectomy. Surg Neurol Int. 2011;2.

 

# 7: on page 10: in nearly 35% of the patients cranioplasty was performed. I believe it would

be more appropriate to include this data in the methods and results sections rather than just

in the discussion.

The content of cranioplasty has been added to the method and result sections.

Methods

“In addition, the results of cranioplasty, including intraoperative findings, GOS, and complications were reviewed, although the limited nature of the data meant that it could not be fully evaluated.”

Results

Cranioplasty

We observed 37 patients (suture duraplasty group: 27, non-suture duraplasty group: 10) who underwent cranioplasty following a decompressive craniectomy. GOS scores did not differ between the two groups (suture vs. non-suture), and patients with severe head injuries had a poorer prognosis. Therefore, we do not believe that watertight suturing necessarily leads to a good prognosis. In the non-suture duraplasty craniectomy group, the cortex was not exposed because of the formation of fibrotic tissue between the scalp flap and dura material (Fig 3). The incidence of complications, including infection after cranioplasty, were similar in the suture duraplasty and non-suture duraplasty groups (3% and 4%, respectively). Therefore, we believe that non-suture duraplasty may not cause any problems in cranioplasty, although further evaluation is warranted.

---

## [Decision Letter · Decision Letter 1]

18 Aug 2020

PONE-D-20-10922R1

Efficacy and safety of decompressive craniectomy with non-suture duraplasty in patients with traumatic brain injury

PLOS ONE

Dear Dr. Yee,

Thank you for submitting your manuscript to PLOS ONE. After careful consideration, we feel that it has merit but does not fully meet PLOS ONE’s publication criteria as it currently stands. Therefore, we invite you to submit a revised version of the manuscript that addresses the points raised during the review process.

ACADEMIC EDITOR: This is a well-written paper. However, the authors should address all the concerns raised by the reviewers.

We look forward to receiving your revised manuscript.

Kind regards,

Giovanni Grasso

Academic Editor

PLOS ONE

Additional Editor Comments (if provided):

This is a well-written paper. However, the authors should address all the concerns raised by the reviewers.

Reviewers' comments:

Reviewer's Responses to Questions

**Comments to the Author**

1. If the authors have adequately addressed your comments raised in a previous round of review and you feel that this manuscript is now acceptable for publication, you may indicate that here to bypass the “Comments to the Author” section, enter your conflict of interest statement in the “Confidential to Editor” section, and submit your "Accept" recommendation.

Reviewer #1: All comments have been addressed

Reviewer #2: All comments have been addressed

2. Is the manuscript technically sound, and do the data support the conclusions?

Reviewer #1: Yes

Reviewer #2: Partly

3. Has the statistical analysis been performed appropriately and rigorously? 

Reviewer #1: Yes

Reviewer #2: Yes

4. Have the authors made all data underlying the findings in their manuscript fully available?

Reviewer #1: Yes

Reviewer #2: Yes

5. Is the manuscript presented in an intelligible fashion and written in standard English?

Reviewer #1: No

Reviewer #2: Yes

6. Review Comments to the Author

Reviewer #1: I thank the authors for the time spent to improve the paper.

There are, however, a few points that need to be improved:

# The incidence of CSF leak is unclear. Were all cases of wound dehiscence associated with CSF fistula? Were there cases of wound dehiscence without CSF fistula?

# In table 1, regarding the diagnosis, I believe that the items "epidural hematoma with contusion" and "intracerebral hematoma" should be merged into a single item: "intracerebral contusion / hematoma". Epidural hematomas do not require decompressive craniectomy.

# Regarding cranioplasty, I think the data would be clearer if it was plotted in a table, similarly to Table 2. Furthermore, it is important to emphasize that the results session should only address study results. Phrases like "Therefore, we do not believe that watertight suturing necessarily leads to a good prognosis." should be reserved for the discussion or conclusions sessions.

# Additionally, the text needs clarification on several points. I believe professional language editing service is needed. There are lots of them available on the internet. Actually, I believe that this is the main issue with the manuscript at the moment.

Reviewer #2: The authors have chosen to evaluate whether water-tight closure contributes significantly to the success of decompressive craniectomy in TBI. This was done as a retrospective analysis examining data over a 23-month period.

The authors have clearly delineated the data acquisition and analysis. Of note, they were inclusive or relevant outcome measures needed to assess meaningful impact.

The results are presented clearly.

The Discussion and Conclusions are sound and supported by the data analysis presented.

Areas of clarification are in the Methodology utilized and concern for unwarranted selection bias. Specifically, the patient cohort selection criteria are not well explained. Why is age >70 years an exclusion criterion. What is the basis for excluding patients undergoing simultaneous surgery?

In sum, the question posed is worthy of investigation and the authors have clearly approached the issue with data analysis and conclusions that are sound and support the conclusions.

Would recommend clarification of the Methodologic choices from the perspective of bias. The manuscript once clarified, might add meaningfully to our existing practice options for craniectomies.

7. PLOS authors have the option to publish the peer review history of their article (what does this mean?). If published, this will include your full peer review and any attached files.

Reviewer #1: **Yes: **Eduardo Vieira

Reviewer #2: No

---

## [Author Response · Author response to Decision Letter 1]

8 Sep 2020

Reviewer 1

# The incidence of CSF leak is unclear. Were all cases of wound dehiscence associated with CSF fistula? Were there cases of wound dehiscence without CSF fistula?

CSF fistula can be defined as CSF leak outside the skin. In all cases of wound dehiscence, wound culture was carried out. If the bacteria were identified, it was diagnosed as SSI, and if the bacteria were not specified, it was diagnosed as wound dehiscence. In cases of wound dehiscence, it was difficult to accurately determine whether wound dehiscence was associated with CSF fistula because most of the medical records were not clearly specified or were described as simple discharge even if specified. We were not sure that the discharge was CSF, and we judged that determining whether the CSF fistula is present could lead to inaccurate results. So, we consisted of two items, wound dehiscence and SSI, for values of complication. 

# In table 1, regarding the diagnosis, I believe that the items "epidural hematoma with contusion" and "intracerebral hematoma" should be merged into a single item: "intracerebral contusion/hematoma". Epidural hematomas do not require decompressive craniectomy.

We agree with your opinion, so we modified it to "intracerebral contusion/hematoma" as you pointed out.

# Regarding cranioplasty, I think the data would be clearer if it was plotted in a table, similarly to Table 2. Furthermore, it is important to emphasize that the results session should only address study results. Phrases like "Therefore, we do not believe that watertight suturing necessarily leads to a good prognosis." should be reserved for the discussion or conclusions sessions.

We made Table 3 of the cranioplasty results. However, because there were a few cases, it was difficult to obtain statistically significant results, so the p-value could not be calculated. Also, the incidence rate (3% vs. 4%) previously described was corrected by incorrectly indicating the incidence rate ratio (3:4) between the two groups.

Cranioplasty

We observed 37 patients (suture duraplasty group: 27, non-suture duraplasty group: 10) who underwent cranioplasty following a decompressive craniectomy. GOS scores did not differ between the two groups. And the incidence of complications, including infection after cranioplasty, were similar in the suture duraplasty and non-suture duraplasty groups (Table 3). In the non-suture duraplasty craniectomy group, the cortex was not exposed because of the formation of fibrotic tissue between the scalp flap and dura material (Fig 3). 

The phrase “Therefore ~” was deleted from the result session, and an additional paragraph was written in the discussion session.

DISCUSSION

Although there were not many cases, the results of cranioplasty showed that there was no significant difference between the two groups in GOS and complications. Therefore, we believe that non-suture duraplasty may not cause any problems in cranioplasty, although further evaluation is warranted.

# Additionally, the text needs clarification on several points. I believe professional language editing service is needed. There are lots of them available on the internet. Actually, I believe that this is the main issue with the manuscript at the moment.

We got a language editing service again. A certificate was also attached.

 

Reviewer 2

# Why is age >70 years an exclusion criterion. What is the basis for excluding patients undergoing simultaneous surgery?

Many studies have been reported that old age is one of the independent predictors of a worse outcome for traumatic brain injury. In comparing the prognosis according to the procedure method during craniectomy, we thought that the variable “age” could influence clinical outcomes such as GOS. Also, there are many cases of brain atrophy in elderly patients, and brain atrophy can affect the evaluation of CT findings such as subdural hygroma. For this reason, we excluded patients over 70 years of age from the study.

Simultaneous surgery is a surgery performed in multiple trauma patients (for example, when laparotomy and craniectomy are performed at the same time due to abdominal organ bleeding and TBI). Simultaneous surgery is very rare, with 2-3 cases a year, and most of the patients die due to DIC or hypotension that resulted from massive bleeding. There were 2 cases of simultaneous surgery during our study period. However, the patients died within one week, the postop brain CT required for the study results was not performed, and SSI was not evaluated, so simultaneous surgery was excluded from the study.

---

## [Decision Letter · Decision Letter 2]

23 Sep 2020

Efficacy and safety of decompressive craniectomy with non-suture duraplasty in patients with traumatic brain injury

PONE-D-20-10922R2

Dear Dr. Yee,

We’re pleased to inform you that your manuscript has been judged scientifically suitable for publication and will be formally accepted for publication once it meets all outstanding technical requirements.

Kind regards,

Giovanni Grasso

Academic Editor

PLOS ONE

Additional Editor Comments (optional):

Reviewers' comments:

Reviewer's Responses to Questions

**Comments to the Author**

1. If the authors have adequately addressed your comments raised in a previous round of review and you feel that this manuscript is now acceptable for publication, you may indicate that here to bypass the “Comments to the Author” section, enter your conflict of interest statement in the “Confidential to Editor” section, and submit your "Accept" recommendation.

Reviewer #1: All comments have been addressed

Reviewer #2: All comments have been addressed

2. Is the manuscript technically sound, and do the data support the conclusions?

Reviewer #1: Yes

Reviewer #2: Yes

3. Has the statistical analysis been performed appropriately and rigorously? 

Reviewer #1: Yes

Reviewer #2: Yes

4. Have the authors made all data underlying the findings in their manuscript fully available?

Reviewer #1: Yes

Reviewer #2: Yes

5. Is the manuscript presented in an intelligible fashion and written in standard English?

Reviewer #1: Yes

Reviewer #2: Yes

6. Review Comments to the Author

Reviewer #1: The authors addressed all our comments and made all the corrections needed. I believe the manuscript is ready for publication.

Reviewer #2: Understood reasons cited for exclusion/inclusion criteria.

However, given the conservative age limits of <70yrs, would add caveat to Conclusions that applicable only to the non-geriatric cohort.

7. PLOS authors have the option to publish the peer review history of their article (what does this mean?). If published, this will include your full peer review and any attached files.

Reviewer #1: No

Reviewer #2: **Yes: **Odette Harris

---

## [Editor Report · Acceptance letter]

29 Sep 2020

PONE-D-20-10922R2 

Efficacy and safety of decompressive craniectomy with non-suture duraplasty in patients with traumatic brain injury 

Dear Dr. Yee:

I'm pleased to inform you that your manuscript has been deemed suitable for publication in PLOS ONE. Congratulations! Your manuscript is now with our production department. 

Kind regards, 

on behalf of

Professor Giovanni Grasso 

Academic Editor

PLOS ONE